# Structure of *Dunaliella* photosystem II reveals conformational flexibility of stacked and unstacked supercomplexes

Ido Caspy[1], Maria Fadeeva[1], Yuval Mazor[2,3]*, Nathan Nelson[1]*

[1]Department of Biochemistry and Molecular Biology, The George S. Wise Faculty of Life Sciences, Tel Aviv University, Tel Aviv, Israel; [2]School of Molecular Sciences, Arizona State University, Tempe, United States; [3]Biodesign Center for Applied Structural Discovery, Arizona State University, Tempe, United States

**Abstract** Photosystem II (PSII) generates an oxidant whose redox potential is high enough to enable water oxidation , a substrate so abundant that it assures a practically unlimited electron source for life on earth . Our knowledge on the mechanism of water photooxidation was greatly advanced by high-resolution structures of prokaryotic PSII . Here, we show high-resolution cryogenic electron microscopy (cryo-EM) structures of eukaryotic PSII from the green alga *Dunaliella salina* at two distinct conformations. The conformers are also present in stacked PSII, exhibiting flexibility that may be relevant to the grana formation in chloroplasts of the green lineage. CP29, one of PSII associated light-harvesting antennae, plays a major role in distinguishing the two conformations of the supercomplex. We also show that the stacked PSII dimer, a form suggested to support the organisation of thylakoid membranes , can appear in many different orientations providing a flexible stacking mechanism for the arrangement of grana stacks in thylakoids. Our findings provide a structural basis for the heterogenous nature of the eukaryotic PSII on multiple levels.

*For correspondence:
ymazor@asu.edu (YM);
nelson@tauex.tau.ac.il (NN)

**Competing interest:** The authors declare that no competing interests exist.

## Editor's evaluation

The study provides exceptional new insights into the structural organization of the light energy capturing machinery of photosynthesis by resolving the structure of the water-splitting photosystem II and associated complexes isolated from the green alga Dunaliella using cryo-electron microscopy and modeling. The results indicate large-scale flexibility of photosystem II – light harvesting complex II supercomplex, which are likely to have functional consequences for the stacking of thylakoid membranes, interactions of complexes into mesoscale structures, which may control the funneling light energy into the photosystems or photoprotective mechanisms, controlling light-driven fluxes of electrons and protons.

## Introduction

In eukaryotes, the light reaction of oxygenic photosynthesis occurs in chloroplasts. Four protein complexes essential for the light reactions reside in an elaborate membrane system of flattened sacs called thylakoids (*Nelson and Ben-Shem, 2004*). From these four complexes, the photosystem II (PSII) complex catalyses light-driven water oxidation and provides the electrons used for carbon fixation (*Vinyard et al., 2013*; *Nelson and Junge, 2015*).

Thylakoids form a physically continuous three-dimensional network, differentiated into two distinct physical domains: cylindrical stacked structures (called grana) and connecting single membrane regions (stroma lamellae). Photosystem I (PSI) is mainly located in the stroma lamellae, while PSII is

found almost exclusively in the grana (*Hankamer et al., 1997*; *Anderson, 2002*; *Kim et al., 2005*). Grana stacking is a dynamic process dependent on the internal osmotic pressure, the luminal ion composition, and environmental cues and is thought to be supported by interactions among PSII complexes (*Rubin et al., 1981*; *Barber et al., 1980*; *Kirchhoff, 2014*; *Liu et al., 2018*; *Dalal and Tripathy, 2018*, *Garab and Mustárdy, 2000*; *Kirchhoff et al., 2007*).

PSII is a homodimer with a molecular mass of ~500 kDa, each monomer contains cofactors such as chlorophylls (Chls), quinones, carotenoids, and lipids which are coordinated by at least 20 protein subunits (*Shen, 2015*; *Cox et al., 2020*; *Barber, 2004*). In each PSII core, a cluster of four manganese (Mn) and one calcium (Ca) carries out $H_2O$ oxidation and $O_2$ release (*Nelson and Yocum, 2006*; *McEvoy and Brudvig, 2006*). The eukaryotic reaction centre is a dimer surrounded by tightly bound monomeric light-harvesting complexes (LHCs) and trimeric LHCII complexes (*Daskalakis, 2018*; *Mascoli et al., 2020*; *Barber et al., 2002*). Two monomeric LHCs, CP26, and CP29 are located between LHCII trimers and PSII core subunits (*Nield et al., 2000*) additional LHCII trimers can bind PSII depending on light intensity and quality (*Goldschmidt-Clermont and Bassi, 2015*).

Although more than 3 billion years of evolution separate cyanobacteria, red algae, green algae and plants, and high-resolution PSII structures show that each PSII monomer along with its dimeric arrangement is highly conserved, especially in the membrane-bound regions of the PSII (*Wei et al., 2016*; *Su et al., 2017*; *Pi et al., 2019*; *van Bezouwen et al., 2017*; *Sheng et al., 2019*; *Ago et al., 2016*). Structural and spectroscopic investigations uncovered various aspects of PSII's water splitting mechanism, but a complete model is still missing (*Umena et al., 2011*; *Cox et al., 2014*; *Suga et al., 2019*; *Kupitz et al., 2014*; *Kern et al., 2018*). Most of the mechanistic and structural studies of PSII were performed in thermophilic cyanobacteria, but structural studies of PSII from the eukaryotic lineage are lagging in terms of resolution and water molecules network (*Ago et al., 2016*; *Sheng et al., 2019*; *Shen et al., 2019*; *Wei et al., 2016*; *Su et al., 2017*; *Pi et al., 2019*; *van Bezouwen et al., 2017*; *Nagao et al., 2019*).

In this work, PSII was isolated from the halotolerant green alga *Dunaliella salina*. A high-resolution (2.43 Å) structure of PSII shows structural properties of the *Dunaliella* PSII supercomplex. The eukaryotic PSII appears to exist in two distinct core conformations that differ substantially in their inner dimer separation and the location of CP29, an important monomeric LHC. Structural analysis of stacked PSII dimers showed highly flexible interactions which can play a role in the dynamic organisation of chloroplast membranes. These findings introduce an additional, underlying, level of organisation, which can impact its excitation energy transfer (EET) properties and the overall organisation of the thylakoid membranes.

## Results and discussion

### Two distinct PSII conformations in green alga

Highly active PSII from *D. salina* cells was applied on glow-discharged holey carbon grids that were vitrified for cryo-EM structural determination (see Methods). Initial classification of the dataset showed that approximately 20% of the particle population were in a stacked PSII configuration, containing two PSII dimers facing each other on their stromal side (*Figure 1—figure supplement 1*). From the unstacked PSII dimers, approximately 20% were in the C2S configuration (two Cores, one Stable LHCII), which was previously identified by low-resolution cryo-EM (*Drop et al., 2014*; *Figure 1—figure supplement 1c*). The majority of the PSII particles contained two LHCII in the C2S2 configuration. The map of the C2S2 particles refined to a global resolution of 2.82 Å (*Figure 1—figure supplement 2*). Close examination of this map revealed that CP29, one of the monomeric LHC proteins, implicated as a junction for EET from LHCII trimers to PSII core, appeared to be in lower resolution than the rest of the supercomplex (*Figure 1—figure supplement 2a*). Indeed, when this particle set was further classified, two distinct PSII conformations of the PSII supercomplex became apparent. In these two conformations, the two PSII cores are shifted laterally with respect to each other (*Figure 1*). This lateral shift is accompanied by several other associated movements, most noticeably, a large movement of the CP29 subunit (*Figure 1*) in line with our initial observation. The two conformers were denoted compact and stretched PSII (C2S2$_{COMP}$ and C2S2$_{STR}$, respectively), and the high-specific activity of 816 µmol $O_2$/(mg Chl * hr) measured for the preparation prior to vitrification suggest that both are highly active. The final reconstruction of the compact orientation refined to an overall resolution of

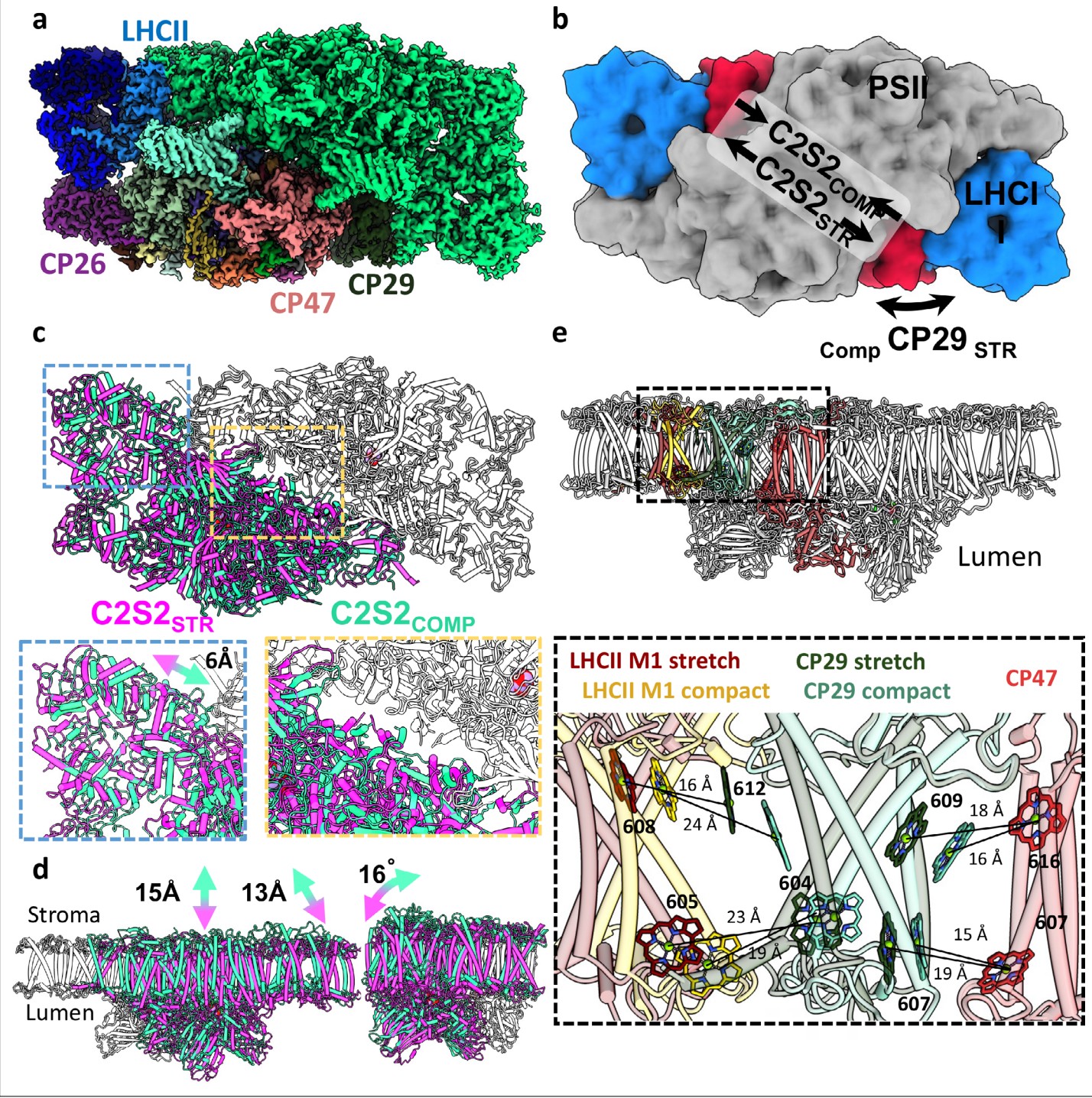

**Figure 1.** Two conformations of the eukaryotic photosystem II (PSII). (**a**) Overall view of the PSII C2S2 map in the compact conformation. One asymmetric unit is coloured in green, and in the other, each chain is coloured individually. PSII is shown from a luminal view in panels a–c. (**b**) Low-resolution model depicting the overall shifts in subunits between the two PSII conformations. CP29 in red, light-harvesting complex II (LHCII) in blue, and the two PSII cores in grey. (**c**) The two PSII conformations were superposed on one asymmetric unit (coloured in grey). The second asymmetric unit is coloured in magenta for the stretched conformation (C2S2$_{STR}$) and green for the compact conformation (C2S2$_{COMP}$). A close up showing a 6 Å shift in the position of LHCII and the lateral displacement between the two cores. (**d**) The stretched PSII conformation shows substantial drop in the membrane plane (13–15 Å, depending on the precise location), contributing to a larger inward curve (compared to the luminal space) of the entire supercomplex. Large deformations in the position of CP26 subunit which rotates by 16° between the two conformations. PSII is shown from a membrane plane view in panels d–e. (**e**) Considerable changes in the position of CP29 affect the transfer rates between LHCII and CP47. CP47 of both conformations (in red) is

*Figure 1 continued on next page*

*Figure 1 continued*

superposed, and distances between key chlorophylls (Chls) of CP29_COMP (light turquoise) and CP29_STR (dark green) show increased transfer distances in the stretched conformation. The distances between LHCII and CP29 follow an opposite trend, decreasing in the stretched conformation (LHCII_STR in dark red) and increasing in the compact conformation (LHCII_COMP in yellow). Distances were measured from the central manganese (Mg) atoms.

The online version of this article includes the following source data and figure supplement(s) for figure 1:

**Figure supplement 1.** Cryo-EM data collection and processing scheme for unstacked and stacked photosystem II (PSII) complexes.

**Figure supplement 2.** Local resolution and Fourier shell correlation (FSC) of *Dunaliella* photosystem II (PSII) supercomplexes.

**Figure supplement 3.** Map densities of unstacked *Dunaliella* photosystem II (PSII).

**Figure supplement 4.** Comparing CP29_STR and CP29_COMP.

**Figure supplement 5.** Comparing CP29 positions between *Dunaliella* and *Chlamydomonas reinhardtii* (*Cr*).

**Figure supplement 6.** Principal component (PC) analysis of C2S2_COMP.

**Figure supplement 7.** Principal component (PC) analysis of C2S2_STR.

**Figure supplement 8.** Changes in CP29 chlorophyll (Chl) positions between C2S2_COMP and C2S2_STR.

**Figure supplement 9.** Sucrose gradient and sodium dodecyl sulfate–polyacrylamide gel electrophoresis (SDS-PAGE) of *Dunaliella* photosystem II (PSII) preparation.

**Figure supplement 9—source data 1.** Uncropped gel image of *Figure 1—figure supplement 9b*.

**Figure supplement 9—source data 2.** Oxygen evolution activity from *Figure 1—figure supplement 9c* in excel format.

---

2.43 Å, the highest of any eukaryotic PSII structures (PDB ID 7PI0; *Figure 1a*; *Supplementary file 1* and *Figure 1—figure supplements 1–3*).

The C2S2_COMP structure is similar to the previously determined C2S2 supercomplex from *Chlamydomonas reinhardtii* or higher plants (*Shen et al., 2019*; *Sheng et al., 2019*) and the cyanobacterial core structures (*Umena et al., 2011*; *Kato et al., 2021*). The second, stretched conformer was solved to 2.62 Å resolution and accounted for about 37% of the unstacked C2S2 PSII particles. *Figure 1* and *Videos 1–2* depict the superposition of the polypeptide chains of the two conformers, showing major differences in the location and orientation of PSII monomers. Superposition of the *Dunaliella* and *Chlamydomonas* C2S2 (PDB 6KAC) structures and maps suggests that the *Chlamydomonas* structure also contains these different conformers. This may explain the decreased local map resolution presented in the aforementioned subunits, compared to the rest of the cryo-EM map (*Sheng et al., 2019*).

## Structures of *D. salina* unstacked PSII at high resolution

Thus far, available PSII structures suggested a single, highly conserved organisation of the two PSII cores (*Su et al., 2017*; *Wei et al., 2016*; *Pi et al., 2019*; *Sheng et al., 2019*; *Umena et al., 2011*;

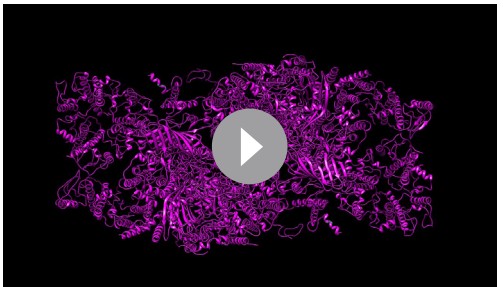

**Video 1.** Different conformations of *Dunaliella* C2S2 photosystem II (PSII). Morph showing the transition from the stretched (magenta) to the compact (green) conformation from a luminal view.

https://elifesciences.org/articles/81150/figures#video1

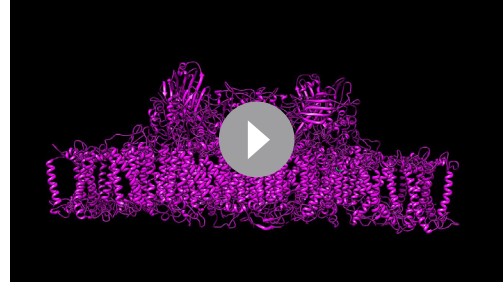

**Video 2.** Different conformations of *Dunaliella* C2S2 photosystem II (PSII). Morph showing the transition from the stretched (magenta) to the compact (green) conformation from a membrane plane view.

https://elifesciences.org/articles/81150/figures#video2

---

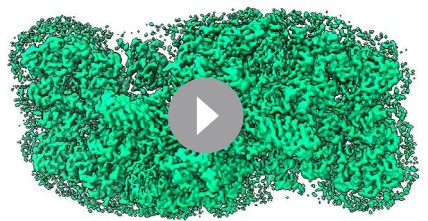

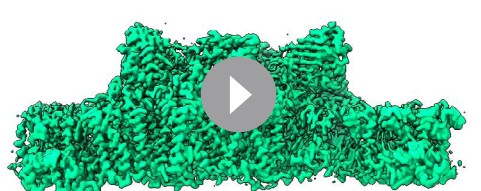

**Video 3.** Continuous heterogeneity in C2S2$_{COMP}$ PC1. Transition between all states in C2S2$_{COMP}$ (green) PC1 from a luminal view, showing photosystem II (PSII) monomers change in location and orientation along the intermonomer space.
https://elifesciences.org/articles/81150/figures#video3

**Video 4.** Continuous heterogeneity in C2S2$_{COMP}$ PC3. Transition between all states in C2S2$_{COMP}$ (green) PC3 from a membrane plane view, showing photosystem II (PSII) monomers change in location and orientation along the membrane plane.
https://elifesciences.org/articles/81150/figures#video4

*Ago et al., 2016*). The high-resolution structures of *Dunaliella* C2S2$_{COMP}$ and C2S2$_{STR}$ provide a new perspective on the dynamic arrangement of eukaryotic PSII and the interaction of the core complex with its LHCs. To compare the C2S2$_{COMP}$ and C2S2$_{STR}$, the core complexes were aligned (*Figure 1* and *Videos 1–2*). Initial inspection showed that one of the major differences between the two conformations is the orientation of CP29 (*Figure 1e*). In C2S2$_{STR}$ CP29, helices A and C move towards the LHCII trimer of the opposite monomer and away from CP47, with helix B of CP29 serving as a rotation axis. Moreover, CP29$_{COMP}$ contained only 9 Chls, compared to 11 in CP29$_{STR}$ and 13 Chl in *Chlamydomonas* PSII CP29 (*Sheng et al., 2019* and *Supplementary file 2*). Chls 605 and 616 were absent in both structures, and CP29$_{COMP}$ was also missing Chls 611 and 613. This might be attributed to the flexibility of CP29 C-terminus (which is proximal to 613), or some side chains and ligands rearrangement associated with the movement of CP29. The structure and b-factor of CP29$_{COMP}$ and CP29$_{STR}$ are similar (*Figure 1—figure supplement 4*); however, superposition of C2S2$_{COMP}$, C2S2$_{STR}$, and *Chlamydomonas* C2S2M2L2 (*Sheng et al., 2019*) suggests that CP29$_{COMP}$ may be an intermediate conformation between CP29$_{STR}$, that is bound to the S-trimer through its C-terminus, and *Chlamydomonas* CP29, that binds the M- and L-trimers via the C-terminus (*Figure 1—figure supplement 5*).

In the C2S2$_{STR}$ conformation, the PSII monomers slid in the membrane plane along the central symmetry axis separating them (*Figure 1b* and *Videos 1–2*). The non-aligned core shows the extent of the shift in the core peptides together with the minor LHCs and LHCII trimer (*Figure 1c and d*). As a result, all the interactions at the core's interface are modified, leading to local changes in chain orientations and the conformations of some loops. Core subunits at the centre of the monomer displayed a greater shift (D1, D2, CP47 CP43, and PsbO were displaced by 6–10 Å; *Figure 1*), and the peripheral subunits showed the largest shift and tilt compared to C2S2$_{COMP}$ (PsbE, PsbP, and CP26 moved by 13 Å, PsbZ showed the largest relocation of nearly 15 Å, and CP26 showed a maximal tilt of 16°; *Figure 1*). Multibody refinement (*Nakane et al., 2018*) of both C2S2$_{COMP}$ and C2S2$_{STR}$ demonstrated that the two PSII monomers in each conformation contain

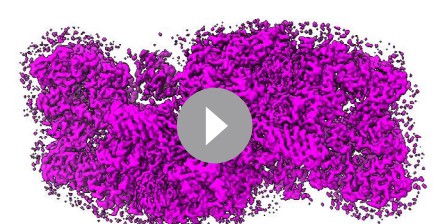

**Video 5.** Continuous heterogeneity in C2S2$_{STR}$ PC1. Transition between all states in C2S2$_{STR}$ (magenta) PC1 from a luminal view, showing photosystem II (PSII) monomers change in location and orientation along the intermonomer space.
https://elifesciences.org/articles/81150/figures#video5

additional structural heterogeneity (*Figure 1—figure supplements 6–7* and *Videos 3–6*).

## Distinct CP29 conformations alter LHCII to PSII core connectivity

The observed conformational change of CP29 alters EET pathways from LHCII to the PSII core and may account for the differences between calculated and measured EET (*Chmeliov et al., 2014*; *Chmeliov et al., 2016*; *Caffarri et al., 2011*; *Mascoli et al., 2020*; *Croce and van Amerongen, 2011*; *Croce and van Amerongen, 2020*; *van der Weij-de Wit et al., 2011*). To assess changes in transfer rates between the stretched and compact orientations, we measured how the distances between the closest Chls of CP47 (PSII core), CP29, and LHCII change between the two PSII conformations (all reported Chl distances are measured from the central Mg atom). Overall, we find that CP29 and LHCII move closer to each other and away from CP47 in the stretched configuration. The average distances between CP29 Chls 603, 607, and 609 to the CP47 Chls 607 and 616, increased from 17 Å to 20 Å between the compact to stretched conformations, suggesting faster transfer rates from CP29 to CP47 in the compact conformation. In contrast to this, the average distances between CP29 Chls 604 and 612 to LHCII Chls 604 and 608 increased from 20 Å in the stretched conformation to 23 Å in the compact conformation, suggesting that transfer from LHCII to CP29 is slower in the compact orientation (*Figure 1e*). The missing CP29 Chls 611 and 613 form part of the interface to LHCII and are missing in the compact conformation, which should also contribute to slower transfer rates from LHCII to CP29 in the compact configuration (*Figure 1—figure supplement 8*). Altogether, transfer from LHCII to the PSII core should be considerably slower in the compact orientation from both distance and Chl occupancy considerations. Similar features of altered Chl conformations were identified in molecular dynamics (MDs) simulation of LHCII exploring its structural dynamics (*Liguori et al., 2015*) compared to its crystal structure (*Liu et al., 2004*). The analysis showed differences in the excitonic coupling of Chl clusters 606–607 and 611–612. MD suggested an increase in the interaction energies of 606–607 and a decrease in the interaction energies of similar proportion in 611–612 (*Liguori et al., 2015*). The 611–612 Chl pair was proposed as a light-harvesting regulator of EET from CP29 to CP47 and as a quenching site (*Caffarri et al., 2011*; *Ruban et al., 2007*; *Novoderezhkin et al., 2005*; *Pascal et al., 2005*), as its change in fluorescence yield was attributed to a protein conformational change that leads to a redistribution of the interpigment energetics (*Valkunas et al., 2012*).

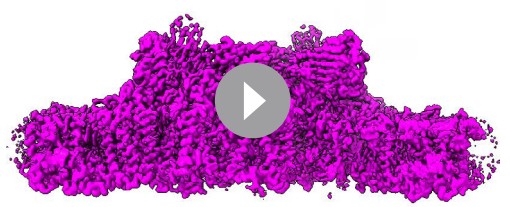

**Video 6.** Continuous heterogeneity in C2S2$_{STR}$ PC2. Transition between all states in C2S2$_{STR}$ (magenta) PC2 from a membrane plane view, showing photosystem II (PSII) monomers change in location and orientation along the membrane plane.
https://elifesciences.org/articles/81150/figures#video6

## The compact and stretched PSII conformations contain substantial levels of continuous structural heterogeneity

Using multibody refinement (*Nakane et al., 2018*), with each PSII monomer defined as a separate rigid body, significantly improved the resolution and map quality in both C2S2$_{COMP}$ and C2S2$_{STR}$, showing that substantial structural heterogeneity exists in both datasets at the level of PSII monomers. Analysing the shape of the heterogeneity in C2S2$_{COMP}$ and C2S2$_{STR}$, using principal component analysis (PCA) showed that the first six principal components (PCs) explain more than 85% of the variance in the data and consist of continuous heterogeneity (*Figure 1—figure supplements 6–7*; *Videos 3–6*). Substantial displacements of approximately 13 Å are observed between the two monomers in the compact conformation (*Figure 1—figure supplement 6*), and a larger range of displacements (up to 20 Å) exists in the stretched conformation (*Figure 1—figure supplement 7*). The direction of PCs describes translations perpendicular and parallel to the membrane plane. This suggests that both conformations are flexible and can respond to different membrane curvature (*Videos 3–6*). To examine the possible effects on energy transfer, we measured the change in intermonomer Chl distances across the different components. As expected, the PCs describing changes in the membrane plane

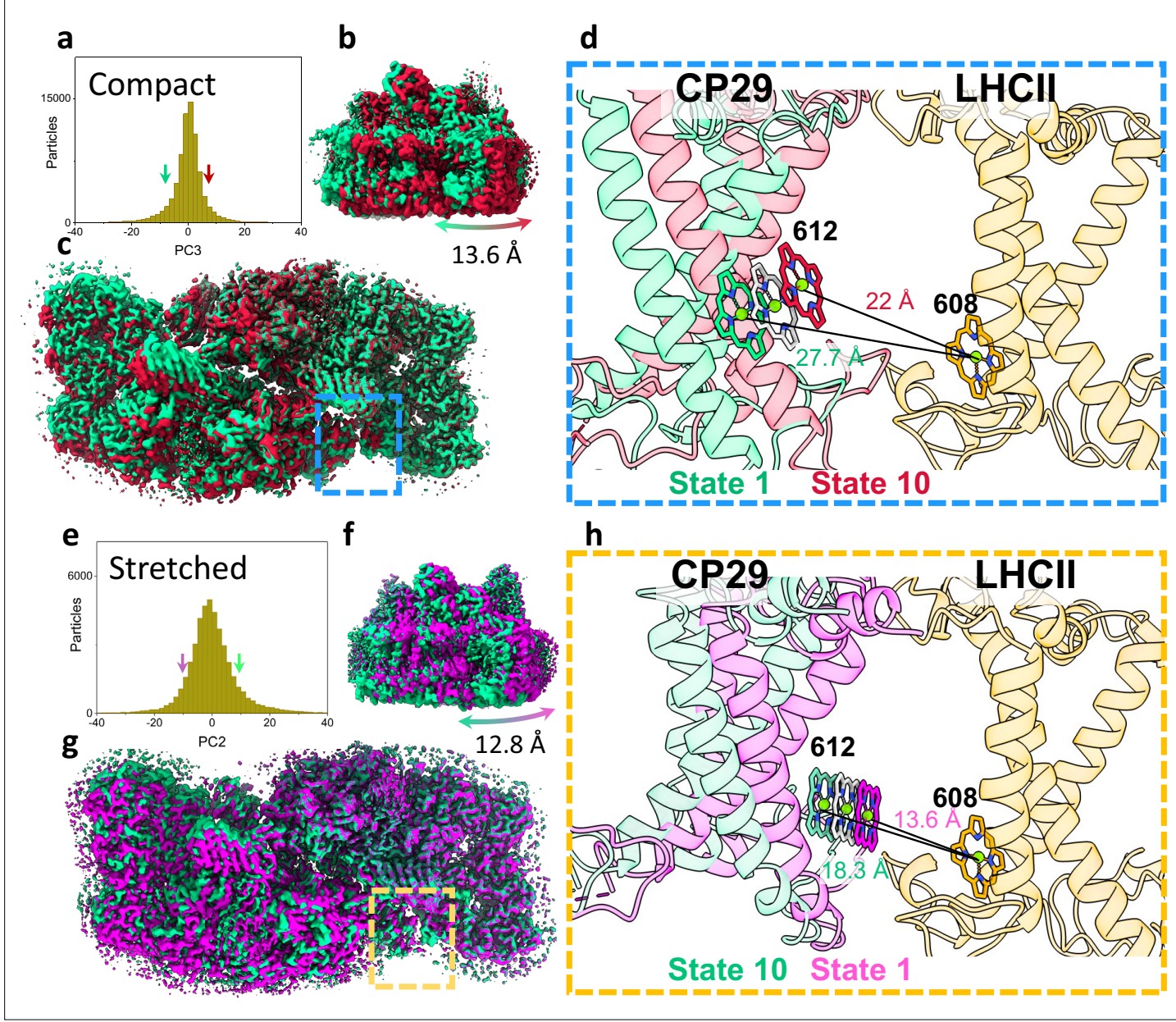

**Figure 2.** Heterogeneity within photosystem II (PSII) states.

(**a**) Continuous heterogeneity of C2S2$_{COMP}$ particles distribution along the third principal component (PC) axis. Each PC was divided into 10 states separated by 9% of the particle population along the PC axis. State 1 (corresponding to the position of the ninth percentile on the PC axis) is marked with a green arrow and state 10 (corresponding to the position of the 91st percentile on the PC axis) with a red arrow. States 1 and 10 are coloured in green and red in panels a–d. (**b**) To maximize the state differences on the left monomer, the maps were superposed along the region of the right PSII monomer. Membrane plane view of the shift in position of C2S2$_{COMP}$ in the third PC. (**c**) Luminal view of the shift in position of C2S2$_{COMP}$. CP29 and light-harvesting complex II (LHCII) are marked with a blue rectangle (**d**) Zoom-in on the change in CP29 position between states 1 and 10. The change in distance between CP29 Chl 612 and LHCII M1 Chl 608 is shown. LHCII M1 (from the superposed PSII monomer) is coloured light orange, and the consensus position of Chl 612 is shown in grey. (**e**) Continuous heterogeneity of C2S2$_{STR}$ particles distribution in the second PC. State 1 is marked with a magenta arrow and state 10 with a teal arrow. Colours are maintained in panels e–h. (**f**) Membrane plane view of the shift in position of C2S2$_{STR}$ in the second PC. (**g**) Luminal view of the shift in position of C2S2$_{STR}$. CP29 and LHCII are marked with an orange rectangle. (**h**) Zoom-in on the change in CP29 position between states 1 and 10. The change in distance between CP29 Chl 612 and LHCII M1 Chl 608 is shown. LHCII M1 is coloured orange, and the consensus position of Chl 612 is shown in grey.

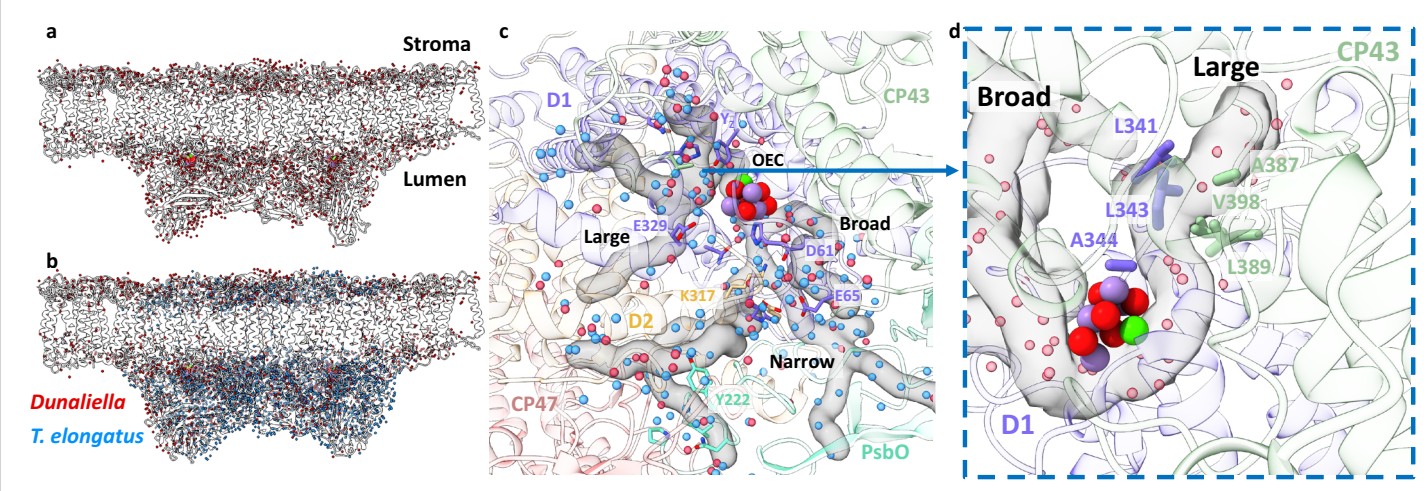

**Figure 3.** Water distribution and channels in eukaryotic photosystem II (PSII).
(a) Water molecules distribution in *Dunaliella* C2S2$_{COMP}$ structure. The protein scaffold is coloured grey, and water molecules are shown as red spheres. (b) Water molecules distribution in *Dunaliella* C2S2$_{COMP}$ compared to *Thermosynechococcus elongatus*. PSII core (PDBID 3WU2). *T. elongatus* water molecules are shown as blue spheres. (c) *Dunaliella* PSII water channels identified by CAVER analysis, shown as grey transparent maps. The Large, Narrow, and Broad channels are annotated along with selected amino acids coloured according to their respective subunits. Water molecules are presented as in panel b. (d) A hydrophobic patch identified in the large channel near the oxygen-evolving complex (OEC), which may serve as an O$_2$ release pathway. The region shown is indicated by the blue arrow, but the orientation is different to improve visualisation.

The online version of this article includes the following figure supplement(s) for figure 3:

**Figure supplement 1.** Sequence alignment of the hydrophobic residues lining the large channel cavity near oxygen-evolving complex (OEC).

**Figure supplement 2.** Na$^+$ ion and post-translational modifications (PTMs) in *Dunaliella* photosystem II (PSII).

markedly change some key distances between LHCII, CP29, and D1 across monomers (*Figure 2*). This means that within each PSII conformation, substantial levels of heterogeneity in transfer rates should be considered. Changes in Chl positions were observed in CP29 Chls linking CP29 to PSII core and those connecting CP29 with LHCII. These Chls moved by an average distance of more than 5 Å, in both conformations (*Supplementary file 3*). This implies that the association between PSII monomers and between PSII cores and LHCs contains a certain degree of freedom which can modulate EET; the entire assembly may be affected by changes in thylakoid membrane properties such as fluidity, composition, and curvature (*Tardy and Havaux, 1997*; *Johnson and Wientjes, 2020*).

## Water channels and post-translational modifications in *Dunaliella* PSII

More than 1700 water molecules were detected in the C2S2$_{COMP}$ model (*Figure 3a–b*), the first detailed water molecules structure for a eukaryotic PSII. Overall, water molecules are clearly excluded from the membrane space in the PSII core, in contrast, the region occupied by LHC's shows a relatively high number of water molecules in the membrane region. This stems from the presence of several conserved charged amino acids in these antennae and is probably important for the inclusion of such hydrophilic residues within the membrane. We used CAVER (*Chovancova et al., 2012*) to analyse the structure of internal cavities around the oxygen-evolving complex (OEC). As expected from the highly conserved environment around the OEC, the water channels identified previously in the high-resolution cyanobacterial core structure (*Suga et al., 2019*; *Kaur et al., 2019*) are clearly visible in the eukaryotic PSII, and overlap with the results of the internal cavity analysis, these are shown in *Figure 3c* and named 'Large,' 'Narrow,' and 'Broad,' following *Kaur et al., 2019*. When analysing the side chains lining the cavities around the OEC, a small hydrophobic patch, highly conserved in prokaryotes and eukaryotes (*Figure 3—figure supplement 1*), was identified at the beginning of the large channel (*Figure 3d*). This hydrophobic element may facilitate O$_2$ release as part of the catalytic cycle (*Figure 3d*).

Several unique map densities were identified during model building, close to the OEC of both configurations a $Na^+$ ion was modelled. This $Na^+$ ion is coordinated by D1-His337, the backbone carbonyls of D1-Glu333, D1-Arg334, D2-Asn350, and a water molecule, in agreement with the recently identified (*Wang et al., 2020*) binding site (*Figure 3—figure supplement 2a–b*). This agrees with several studies showing that $Na^+$ ions are required for optimal activity of PSII (*Wang et al., 2020*; *Pogoryelov et al., 2003*). Two additional densities, unique to C2S2$_{COMP}$, were observed close to CP29-Ser84 and CP47-Cys218 in the stromal interface between CP29, CP47, and PsbH and within 10 Å of each other. These were modelled as post-translational modifications (PTMs) – Ser84 appears to be phosphorylated and Cys218 seems to be sulfinylated (*Figure 3—figure supplement 2c–d*). Thus far, PTMs were structurally seen in photosystems only as phosphorylated LHCII bound to PSI during state transition (*Pan et al., 2018*; *Huang et al., 2021*; *Pan et al., 2021*). Although they were not identified in-situ, several phosphorylation sites were shown to exist in CP29 large stromal loop (*Chen et al., 2013*; *Liu et al., 2009*; *Poudyal et al., 2020*; *Hansson and Vener, 2003*). CP29 phosphorylation was suggested to be linked with various stress responses, photosynthetic protein degradation, and state transition. Cysteine sulfinylation was shown to be linked to superoxide radical ($O_2.^-$) accumulation, which is subsequently converted by superoxide dismutase to hydrogen peroxide ($H_2O_2$) molecules (*Sevilla et al., 2015*; *Rey et al., 2007*; *Matamoros and Becana, 2021*). CP47-Cys218 is positioned on the outer edge of PSII, close to the stromal end of the thylakoid membrane, and thus is susceptible to oxidation by $H_2O_2$. The map density around Cys218 suggests two cysteine oxidation events which result in the formation of sulfinic acid ($RS-O_2H$).

To summarize, the high-resolution structure of the eukaryotic PSII revealed two distinct states of the PSII complex, adding a new dimension to the known, large compositional heterogeneity of this important system (*Croce and van Amerongen, 2011*; *Caffarri et al., 2009*; *Kouřil et al., 2020*). The increased map resolution resulted in the identification of PTM's, and several conserved hydrophobic residues near the OEC, which may serve as a pathway for the release of $O_2$. In addition to the two distinct conformations, large levels of continuous structural heterogeneity were discovered within each individual state. Multibody analysis (*Nakane et al., 2018*) inherently treats the data as a collection of rigid bodies. This is a good approximation of the heterogeneity in photosynthetic systems but should be regarded as a conservative estimation to additional modes of heterogeneity which exist in this system within each body (*Liguori et al., 2015*).

## The structure of *D. salina* stacked PSII at high resolution

The thylakoid membrane is made of two spatially distinct regions, stroma lamellae and grana stacks, each serving a different role in the photosynthetic process (*Pribil et al., 2014*; *Koochak et al., 2019*). Grana stacks size and numbers are affected by light intensity and ionic composition and can change rapidly (*Wood et al., 2019*). Membrane stacking depends on the presence of cations, mainly $Mg^{2+}$, which is abundant in the thylakoid stroma (*Ishijima et al., 2003*), and between stacked PSII-LHCII (*Wood et al., 2019*). In vitro, suspending chloroplast membranes in low-salt medium cause grana unstacking, and addition of $MgCl_2$ reverts the membranes back to their stacked organisation (*Izawa and Good, 1966*; *Staehelin, 1976*). Several low-resolution cryo-EM models of stacked PSII were obtained in recent years (*Levitan et al., 2019*; *Grinzato et al., 2020*; *Albanese et al., 2017*); these studies were also supplemented by mass spectrometry analysis detecting cross-linked regions across the stroma (*Albanese et al., 2020*), but a high-quality PSII structure that can shed light on the contribution of the supercomplex to thylakoid membrane stacking is missing.

The stacked PSII dataset refined to a 3.68 Å map after applying multibody refinement, with each dimer defined as a rigid body. Subsequently, the stacked particles were classified according to the higher quality PSII dimer, and two distinct populations of stacked PSII dimers were obtained, as observed for the unstacked PSII: one in the C2S2$_{COMP}$ conformation solved to 3.36 Å, and the other in the C2S2$_{STR}$ conformation solved to 3.84 Å (*Figure 4*; *Figure 4—figure supplements 1 and 2*). In both classes, the compact conformation exhibited the best fit for the second, lower resolution, PSII dimer.

Roughly 20 Å separate the two stacked dimers in both classes, as previously shown (*Albanese et al., 2017*). In several regions, this value decreases to approximately 10 Å (*Figure 4d and I*), owing to PSII stromal loops in core subunits and LHCs protruding into the space between the two dimers. Both PSII dimers are shifted by approximately 20 Å relative to each other rather than being perfectly aligned (*Figure 4a*; *Figure 4—figure supplement 3a*). Back projecting the stacked PSII onto an in-vivo

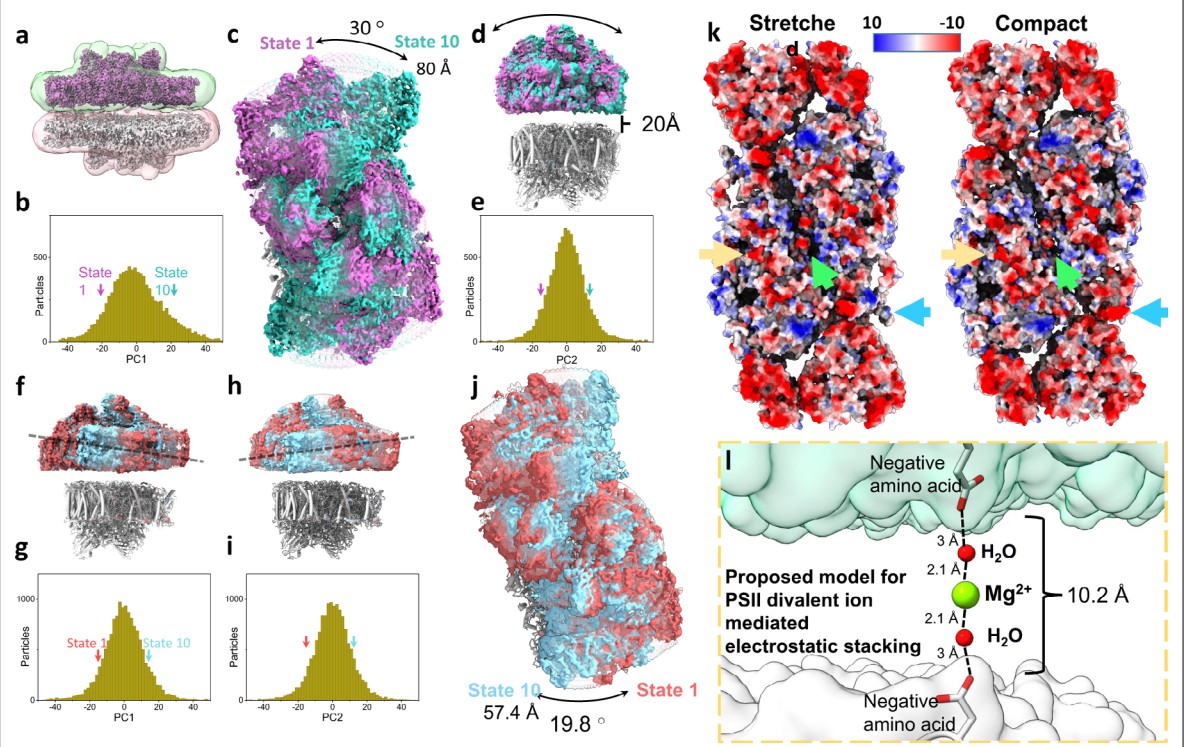

**Figure 4.** Heterogeneity, electrostatic interactions, and model for photosystem II (PSII) stacking.
(**a**) Stacked *Dunaliella* PSII C2S2$_{COMP}$ maps, and the masks used for multibody refinement. Maps are coloured magenta and grey and masks in green and red. (**b**) The particle distribution along the first principal component (PC) shows continuous heterogeneity in the stacked C2S2$_{COMP}$. State 1 is marked with a magenta arrow and state 10 with a teal arrow (colours are preserved in panels b–e). (**c**) Luminal view of the rotation of the upper PSII dimer between state 1 and state 10 (the bottom dimer was kept in a fixed position). (**d**) Membrane plane view of the shift in position of the upper PSII dimer in C2S2$_{COMP}$ second PC. The distance between the upper and lower PSII is shown. (**e**) Particle distribution along the second PC of the stacked C2S2$_{COMP}$ shows continuous heterogeneity. (**f**) Membrane plane view of the tilt in the upper PSII dimer in C2S2$_{STR}$ particle set first PC. The direction of the tilt is marked with a dashed line. State 1 is coloured red and state 10 in cyan (colours are preserved in panels f–j). (**g**) Continuous heterogeneity in the stacked C2S2$_{STR}$ particles distribution along the first PC. (**h**) Membrane plane view of the tilt in orientation of the upper PSII dimer in C2S2$_{STR}$ second PC (with the bottom dimer kept fixed). The dashed line shows the tilt axis is opposite to that shown in panel f. (**i**) Continuous heterogeneity of stacked C2S2$_{STR}$ particles distribution in the second PC. (**j**) Luminal view of the rotation of the upper PSII dimer between state 1 and state 10. (**k**) Coulombic electrostatic potential of the stromal region of the stretched (left) and compact (right) conformations. Differences are marked for CP47 C-terminus (orange arrow), the intermonomer space (green arrow) and CP29 (blue arrow). The negative potentials ($0\ k_BT/e > \Phi > -10\ k_BT/e$) are coloured red, and the positive potentials ($0\ k_BT/e < \Phi < 10\ k_BT/e$) are coloured blue. (**l**) Proposed hypothetical model for PSII stacking mediated by negatively charged amino acids and Mg$^{2+}$ ions (density for Mg$^{2+}$ ions is not observed in our map). Upper PSII shown as a green surface and the lower PSII as a white surface.

The online version of this article includes the following figure supplement(s) for figure 4:

**Figure supplement 1.** Principal component (PC) analysis of stacked C2S2$_{COMP}$.

**Figure supplement 2.** Principal component (PC) analysis of stacked C2S2$_{STR}$.

**Figure supplement 3.** Stromal interactions between photosystem II (PSII) subunits in the stacked configurations.

**Figure supplement 4.** Similar membrane separation in purified and in-vivo detected stacked photosystem II (PSII) complexes.

**Figure supplement 5.** Light-harvesting complex (LHC) interactions limit stacked photosystem II (PSII) rotation.

observed stacked PSII shows that the dimensions of the purified stacked PSII closely match the inter-membrane separation observed in vivo (*Wietrzynski et al., 2020*; *Figure 4—figure supplement 4*).

Altogether in the stacked PSII structure we do not observe any direct protein – protein interactions, this includes loops extending across the stromal gap, this contrasts with previous suggestions (*Albanese et al., 2017*) but can also stem from the absence of loosely associated PSII subunits (specifically PsbR). Below we discuss the extremely flexible nature of the stacked PSII dimer as revealed by

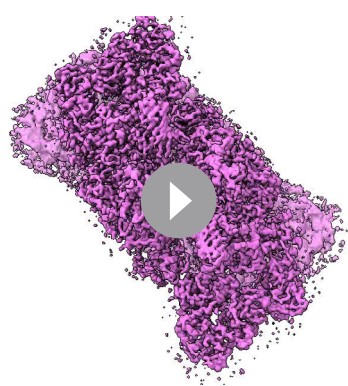

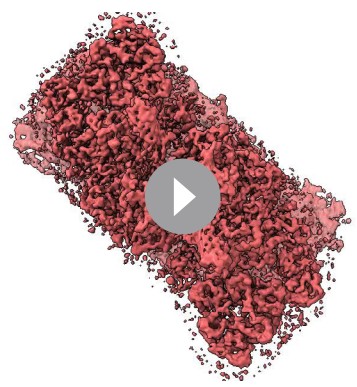

**Video 7.** Continuous heterogeneity in stacked C2S2_COMP PC1. Transition between all states in stacked C2S2_COMP (orchid) PC1 from a luminal view, showing the rotation of the upper photosystem II (PSII) dimer compared to the lower dimer.
https://elifesciences.org/articles/81150/figures#video7

**Video 9.** Continuous heterogeneity in stacked C2S2_STR PC1. Transition between all states in stacked C2S2_STR (red) PC1 from a luminal view, showing the rotation of the upper photosystem II (PSII) dimer compared to the lower dimer.
https://elifesciences.org/articles/81150/figures#video9

multibody analysis. This is consistent with cross-linking results (*Albanese et al., 2020*) and strongly argues against direct protein – protein interactions across the stromal gap.

PCA showed extensive displacements and rotations across the population with stacked PSII dimers rotating relatively to their opposite dimer by as much as 30° and shifting by 80 Å in C2S2_COMP, while in C2S2_STR, the rotation is more restricted, showing a maximum of 19.8° and a shift of 57 Å (*Figure 4* and *Videos 7–10*). The rotation axis of C2S2_COMP appears to be broad region containing the N-termini stromal loops of D2 and CP29 on one dimer, and the stromal loop connecting D2 helices IV and V, CP43 N-terminus, and the C-termini of CP43, CP47, and PsbI on the opposite dimer (*Figure 4—figure supplement 5*). In the stacked C2S2_STR, these stacking interactions also include a stromal loop from D1 which is pushed in the stromal gap by a change in the position of the PsbT C-terminus (green arrow in *Figure 4k*), this shift pushes this D1 loop (connecting helices IV and V) into the stromal space and closer to the adjacent dimer (*Figure 4—figure supplement 3c*). On the axis of rotation which consists of PSII core subunits, additional interactions between different LHCs seem to be essential to maintain stacking. All the rotation states include some degree of LHCs interaction across the stromal gap between opposite PSII dimers, and these seem to limit the extent of possible rotational states. In the stacked C2S2_COMP particle set, the larger range of rotations means that at the extreme states

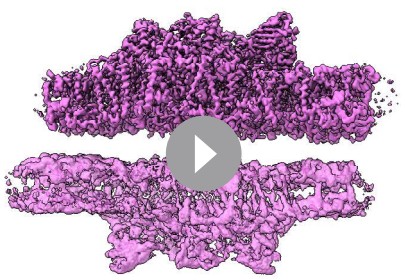

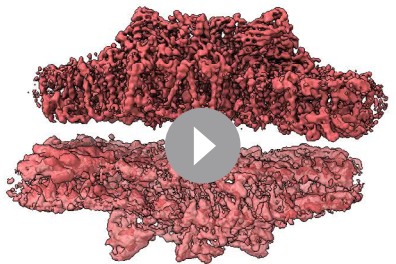

**Video 8.** Continuous heterogeneity in stacked C2S2_COMP PC2. Transition between all states in stacked C2S2_COMP (orchid) PC2 from a membrane plane view, showing upper photosystem II (PSII) dimer tilting to and from the lower dimer.
https://elifesciences.org/articles/81150/figures#video8

**Video 10.** Continuous heterogeneity in stacked C2S2_STR PC2. Transition between all states in stacked C2S2_STR (red) PC2 from a membrane plane view, showing upper photosystem II (PSII) dimer tilting to and from the lower dimer.
https://elifesciences.org/articles/81150/figures#video10

CP26 and LHCII M2 are not involved in stacking interaction and can pair with additional complexes (*Figure 4—figure supplement 5a*), while in the stacked C2S2$_{STR}$ particle set, the smaller rotational range seem to be restricted by CP26 and LHCII M2 interactions (*Figure 4—figure supplement 5b*). These differences, when repeated over many stacked complexes (with additional LHCII complexes), can translate into substantial changes in thylakoid membrane stacking (*Yakushevska et al., 2003*).

The Mg to Mg distances between Chls in each of the stacked complexes (all above 50 Å) make EET between them inefficient. The closest protein contacts are found at the interface between core subunits from both PSII dimers and CP29, supported by peripheral interactions between LHCII trimer and CP26. Most of the PSII stromal surface is electronegative, and accordingly, most of the amino acids that seem to be involved in stacking interactions are either negatively charged or uncharged (*Figure 4k*). Interactions spanning 10 Å are probably insufficient to maintain PSII in its stacked arrangement; however, if mediated by a Mg$^{2+}$ ion and two-to-four H$_2$O molecules, stacking can be stabilized (*Figure 4l*). These interactions comply with the large degree of rotational freedom observed in the stacked dimers and with the strong dependance of stacked dimers on the presence of Mg$^{2+}$ ions and may contribute to thylakoid membrane stacking (*Staehelin, 1976*). Indeed, a cation current counter-acting the positive charges of the proton influx during light is known to occur in chloroplasts (*Hind et al., 1974*; *Nami et al., 2021*; *Barber, 1980*; *Kaňa, 2016*; *Li et al., 2021*; *Kirchhoff et al., 2004*; *Puthiyaveetil et al., 2017*). This has been suggested as the basis for some light-dependent alteration in the stromal spacing of thylakoid membranes (*Puthiyaveetil et al., 2014*; *Kirchhoff et al., 2011*). We suggest that the stacked PSII structure (which strongly depends on the presence of cations during purification) only relies on these weak interactions for its formation and is inherently extremely flexible in all dimensions but the PSII dimer separation distance. This flexibility may explain why stacked PSII structures are rarely detected in-vivo (*Wietrzynski et al., 2020*). However, when stacked PSII structures were detected using cryo-electron tomography, the identified configuration closely matched the stacked PSII dimer identified in this work (*Wietrzynski et al., 2020*); it is possible that in the native membrane state, range of motions in the stacked dimer or that the population of the extreme states increases, leading to the larger variability observed in vivo.

## Summary

The structure of PSII from *Dunaliella* revealed an unexpected level of conformational flexibility in this highly conserved system. The two stable conformations appear to differ in their antennae connectivity and should be considered in PSII modelling attempts. Within each state, the large degree of structural heterogeneity also contributes to EET and may facilitate transitioning between the different states. In the stacked PSII dimer, we do not find any evidence for direct protein interaction connecting the two stacked systems, instead, long range electrostatic interaction between the core PSII subunits are flexible enough to allow for a wide range of motion, and their dependance on cation concentration provides a basis for light dependent regulation (*Puthiyaveetil et al., 2017*; *Hind et al., 1974*).

## Methods

### *Dunaliella* PSII sample preparation

*D. salina* (strain CONC-007) cells were cultured in a 10 l BG11 medium, supplemented with 1.5 M NaCl, 6 μg/ml ferric ammonium citrate, and 50 mM NaHCO$_3$ at pH 8 (*Caspy et al., 2020*). The cells were grown with constant stirring and air bubbling under continuous white light (70 μE) at 25°C for 1 week. After reaching an OD$_{730}$ of 0.4, the culture was harvested by centrifugation at 4000 g for 10 min and resuspended in a medium containing 50 mM HEPES pH 7.5, 300 mM sucrose, and 5 mM MgCl$_2$. The cells were washed once in the same buffer and suspended in a buffer containing 25 mM MES, pH 6.5, 10 mM CaCl$_2$, 10 mM MgCl$_2$, 1 M betaine, 5 mM EDTA, and 12.5% glycerol. Protease-inhibitors cocktail was added to give final concentrations of 1 mM phenylmethylsulfonyl fluoride (PMSF), 1 μM pepstatin, 60 μM bestatin, and 1 mM benzamidine (*Tokutsu et al., 2012*). The cells were disrupted by an Avestin EmulsiFlex-C3 at 1500 psi. Unbroken cells and starch granules were removed by centrifugation at 5000 g for 5 min, and the membranes in the supernatant were precipitated by centrifugation in Ti70 rotor at 181,000 g for 1 hr. The pellet was suspended in a buffer containing 25 mM MES, pH 6.5, 10 mM CaCl$_2$, 10 mM MgCl$_2$, 1 M betaine, 5 mM EDTA, and 12.5% glycerol giving a Chl concentration of 0.4 mg/ml. n-Decyl-α-D-Maltopyranoside (α-DM) was added to a final concentration of

1%, and following stirring for 30 min at 4°C, the insoluble material was removed by centrifugation at 10,000 g for 5 min (*Tokutsu et al., 2012*). Supernatant was concentrated by centrifugation in TI-75 rotor at 377,000 g for 80 min. The pellet was suspended in the above buffer containing 0.3% α-DM at Chl concentration of about 1 mg/ml, loaded on sucrose gradients of 10 to 50% in SW-60 rotor and run at 336,000 g for 15 hr. *Figure 1—figure supplement 9a* shows the distribution of green bands in the tubes. The band containing PSII was concentrated by centrifugation at 550,000 g for 2 hr, and the pellet was suspended in a buffer containing 25 mM MES (pH 6.5), 1 mM $CaCl_2$, 5 mM $MgCl_2$, and 0.1% α-DM to give a Chl concentration of 2 mg Chl/ml. SDS-PAGE of the three bands is presented in *Figure 1—figure supplement 9b*. The final preparation exhibited oxygen evolution activity of 816 µmol $O_2$/(mg Chl * hr) under 560 µmol photons * $m^{-2}$ * $s^{-1}$ illumination (*Figure 1—figure supplement 9c*).

## Cryo-EM data collection and processing

Concentrated PSII solution (3 µl) was applied on glow-discharged holey carbon grids (Cu Quantifoil R1.2/1.3) that were vitrified for cryo-EM structural determination using a Vitrobot FEI (3 s blot at 4°C and 100% humidity). The images were collected using a 300 kV FEI Titan Krios electron microscope, with a slit width of 20 eV on a GIF-Quantum energy filter, at the EMBL cryo facility, Heidelberg, Germany. A Gatan Quantum K3-Summit detector was used in counting mode at a magnification of 130,000 (yielding a pixel size of 0.64 Å), with a total dose of 51.81 e $Å^{-2}$. EPU was used to collect a total of 13,586 images, which were dose-fractionated into 40 movies frames, with defocus values of 0.8–1.9 µm at increments of 0.1 µm. The collected micrographs were motion-corrected and dose-weighted using MotionCor2 (*Zheng et al., 2017*). The contrast transfer function parameters were estimated using CtfFind v.4.1 (*Rohou and Grigorieff, 2015*). A total of 401,467 particles were picked using LoG reference-free picking in RELION3.1 (*Zivanov et al., 2018*). The picked particles were processed for reference-free two-dimensional (2D) averaging. After several rounds of 2D classification, which resulted in 253,804 particles, two initial models was generated using RELION3.1 (*Zivanov et al., 2018*), for the unstacked and stacked PSII.

3D classification of the unstacked PSII revealed two organisations of the LHCs surrounding the core complex – C2S and C2S2. C2S contained 21,066 particles were resampled at a pixel size of 0.896 Å, pooled together, and processed for 3D homogeneous refinement and multibody refinement (*Nakane et al., 2018*) using RELION3.1 (*Zivanov et al., 2018*), giving a final resolution of 3.61 Å. The C2S2 configuration was composed of 75,904 particles with a C2 symmetry, and these were resampled at a pixel size of 0.896 Å, pooled together, and processed for 3D homogeneous refinement and post-processing using RELION3.1 (*Zivanov et al., 2018*), giving a final resolution of 2.82 Å. In an attempt to improve the map density of C2S2, mainly in the vicinity of CP29 and LHCII trimer, 3D classification without refinement was performed and revealed two distinct C2S2 conformations – compact (C2S2$_{COMP}$) and stretched (C2S2$_{STR}$). C2S2$_{COMP}$ was composed of 39,357 particles that undergone symmetry expansion, 3D homogeneous refinement, and multibody refinement (*Nakane et al., 2018*) in C1 symmetry to give a final resolution of 2.43 Å, and C2S2$_{STR}$ was composed of 23,014 particles that undergone symmetry expansion, 3D homogeneous refinement, and multibody refinement (*Nakane et al., 2018*) in C1 symmetry to give a final resolution of 2.62 Å.

23,874 particles that were assigned to the stacked PSII arrangement were resampled at a pixel size of 0.96 Å, pooled together, and processed for 3D homogeneous refinement and multibody refinement (*Nakane et al., 2018*) in C1 symmetry using RELION3.1 (*Zivanov et al., 2018*) and yielded a final resolution of 3.68 Å.

Focused refinement on each individual PSII complex yielded similar resolutions before multibody refinement (3.53 Å and 3.58 Å on each complex), showing both positions are occupied roughly by the same number of complexes. Focused classification was carried out on the upper dimer of the stacked PSII particles to determine if the compact and stretched conformations were also present in the stacked PSII arrangement. This analysis showed that the stacked PSII also contained a mixed population of the compact and stretched conformations. The compact set was composed of 9,567 particles, and these were pooled together and processed for 3D homogeneous refinement followed by multibody refinement (*Nakane et al., 2018*) to give a final resolution of 3.36 Å. The stretched set composed of 14,307 particles, these were pooled together and processed for 3D homogeneous refinement followed by multibody refinement (*Nakane et al., 2018*) to give a final resolution of 3.84 Å. Performing focused

refinement on the lower PSII dimer of both conformations suggested a conformation mixture as well but was less conclusive due to the lower map quality of the lower PSII dimer, and both were fitted with the PSII$_{COMP}$ model (using rigid body refinement) which gave the best overall fit to the map. All the reported resolutions were based on a gold-standard refinement, applying the 0.143 criterion on the Fourier shell correlation between the reconstructed half-maps. (*Figure 1—figure supplement 2*).

PCA was performed using relion_flex_analyse as detailed in *Nakane et al., 2018*. In short, the differences between each particle alignment parameters following the convergence of multibody refinement (at this point, each particle is aligned differently, optimally for each rigid body) are used to represent the heterogeneity in dataset. To generate state maps, the final, consensus, maps of each rigid body are translated along the PC axis to a position corresponding to the stated fraction of the particle population and then added together to generate the specific state map.

## Maps

Focused maps obtained after multibody refinement were combined using the phenix_combined_focused_maps tool (*Liebschner et al., 2019*). Complete models were first refined into the consensus maps and then used to define the combined part of each map, per combined_focused_maps instructions.

## Model building

To generate the C2S2 PSII, the cryo-EM structure of the C2S2 *C. reinhardtii* PSII model PDB 6KAC (*Sheng et al., 2019*) was selected. This model was fitted onto the cryo-EM density map using phenix.dock_in_map in the PHENIX suite (*Liebschner et al., 2019*) and manually rebuilt using Coot (*Emsley et al., 2010*). Stereochemical refinement was performed using phenix.real_space_refine in the PHENIX suite (*Liebschner et al., 2019*). The final model was validated using MolProbity (*Chen et al., 2010*). The refinement statistics are provided in *Supplementary file 1*. Local resolution was determined using ResMap (*Kucukelbir et al., 2014*), and the figures were generated using UCSF Chimera (*Pettersen et al., 2004*) and UCSF ChimeraX (*Goddard et al., 2018*). Representative cryo-EM densities are shown in *Figure 1—figure supplement 3*.

# Acknowledgements

Dr Yael Levi-Kalisman is gratefully acknowledged and thanked for vitrifying the samples. We also thank the Electron Microscopy Core Facility (EMCF) at the European Molecular Biology Laboratory (EMBL) for their support and Felix Weis for data collection and excellent technical support. Molecular graphics and analyses were performed with UCSF Chimera, developed by the Resource for Biocomputing, Visualisation, and Informatics at the University of California, San Francisco, with support from NIH P41-GM103311. Molecular graphics and analyses performed with UCSF ChimeraX, developed by the Resource for Biocomputing, Visualisation, and Informatics at the University of California, San Francisco, with support from National Institutes of Health R01-GM129325 and the Office of Cyber Infrastructure and Computational Biology, National Institute of Allergy and Infectious Diseases. This work was supported by The Israel Science Foundation (Grants No. 569/17 and 199/21), and by German-Israeli Foundation for Scientific Research and Development (GIF), Grant no. G-1483-207/2018. Y.M acknowledges the support by the National Science Foundation under Award No. 2034021.

# Additional information

## Funding

| Funder | Grant reference number | Author |
| --- | --- | --- |
| Israel Science Foundation | 569/17 | Nathan Nelson |
| Israel Science Foundation | 199/21 | Nathan Nelson |
| German-Israeli Foundation for Scientific Research and Development | G-1483-207/2018 | Nathan Nelson |

| Funder | Grant reference number | Author |
|---|---|---|
| National Science Foundation | 2034021 | Yuval Mazor |

The funders had no role in study design, data collection and interpretation, or the decision to submit the work for publication.

## Author contributions

Ido Caspy, Data curation, Software, Formal analysis, Validation, Investigation, Visualization, Methodology, Writing - original draft, Writing - review and editing; Maria Fadeeva, Resources, Investigation, Methodology; Yuval Mazor, Conceptualization, Software, Formal analysis, Supervision, Validation, Visualization, Writing - original draft, Writing - review and editing; Nathan Nelson, Conceptualization, Supervision, Funding acquisition, Validation, Investigation, Visualization, Methodology, Writing - original draft, Project administration, Writing - review and editing

### Author ORCIDs
Yuval Mazor http://orcid.org/0000-0001-5072-0928
Nathan Nelson http://orcid.org/0000-0003-3588-7265

### Decision letter and Author response
Decision letter https://doi.org/10.7554/eLife.81150.sa1
Author response https://doi.org/10.7554/eLife.81150.sa2

## Additional files

### Supplementary files
• Supplementary file 1. Cryo-EM data collection, refinement, and validation statistics.

• Supplementary file 2. CP29 chlorophyll composition determined in previous work.

• Supplementary file 3. Changes in location of CP29 chlorophylls in the compact and stretched conformations principal components (PCs). Shift in chlorophyll position between states 1 and 10 in the first three PCs of the compact (Comp) and stretched (Str) unstacked PSII. Distances are in Å. The average shift and SD are presented for each component.

• MDAR checklist

### Data availability
The atomic coordinates have been deposited in the Protein Data Bank, with accession code 7PI0 (C2S2 COMP ), 7PI5 (C2S2 STR), 7PNK (C2S), 7PIN (stacked C2S2 COMP ) and 7PIW (stacked C2S2 STR ). The cryo-EM maps have been deposited in the Electron Microscopy Data Bank, with accession codes EMD-13429 (C2S2 COMP ), EMD-13430 (C2S2 STR ), EMD-13548 (C2S), EMD-13444 (stacked C2S2 COMP ) and EMD-13455 (stacked C2S2 STR ).

The following datasets were generated:

| Author(s) | Year | Dataset title | Dataset URL | Database and Identifier |
|---|---|---|---|---|
| Caspy I, Fadeeva M, Mazor Y, Nelson N | 2022 | Unstacked compact Dunaliella PSII | https://www.rcsb.org/structure/7PI0 | RCSB Protein Data Bank, 7PI0 |
| Caspy I, Fadeeva M, Mazor Y, Nelson N | 2022 | Unstacked compact Dunaliella PSII | https://www.ebi.ac.uk/emdb/EMD-13429 | EMDB, EMD-13429 |
| Caspy I, Fadeeva M, Mazor Y, Nelson N | 2022 | Unstacked stretched Dunaliella PSII | https://www.rcsb.org/structure/7PI5 | RCSB Protein Data Bank, 7PI5 |
| Caspy I, Fadeeva M, Mazor Y, Nelson N | 2022 | Unstacked stretched Dunaliella PSII | https://www.ebi.ac.uk/emdb/EMD-13430 | EMDB, EMD-13430 |
| Caspy I, Fadeeva M, Mazor Y, Nelson N | 2022 | Unstacked compact Dunaliella PSII | https://www.rcsb.org/structure/7PNK | RCSB Protein Data Bank, 7PNK |

*Continued*

| Author(s) | Year | Dataset title | Dataset URL | Database and Identifier |
|---|---|---|---|---|
| Caspy I, Fadeeva M, Mazor Y, Nelson N | 2022 | Unstacked compact Dunaliella PSII | https://www.ebi.ac.uk/emdb/EMD-13548 | EMDB, EMD-13548 |
| Caspy I, Fadeeva M, Mazor Y, Nelson N | 2022 | Stacked compact Dunaliella PSII | https://www.rcsb.org/structure/7PIN | RCSB Protein Data Bank, 7PIN |
| Caspy I, Fadeeva M, Mazor Y, Nelson N | 2022 | Stacked compact Dunaliella PSII | https://www.ebi.ac.uk/emdb/EMD-13444 | EMDB, EMD-13444 |
| Caspy I, Fadeeva M, Mazor Y, Nelson N | 2022 | Stacked stretched Dunaliella PSII | https://www.rcsb.org/structure/7PIW | RCSB Protein Data Bank, 7PIW |
| Caspy I, Fadeeva M, Mazor Y, Nelson N | 2022 | Stacked stretched Dunaliella PSII | https://www.ebi.ac.uk/emdb/EMD-13455 | EMDB, EMD-13455 |

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
