## [Editor Report]

The study provides exceptional new insights into the structural organization of the light energy capturing machinery of photosynthesis by resolving the structure of the water-splitting photosystem II and associated complexes isolated from the green alga Dunaliella using cryo-electron microscopy and modeling. The results indicate large-scale flexibility of photosystem II – light harvesting complex II supercomplex, which are likely to have functional consequences for the stacking of thylakoid membranes, interactions of complexes into mesoscale structures, which may control the funneling light energy into the photosystems or photoprotective mechanisms, controlling light-driven fluxes of electrons and protons.

---

## [Decision Letter]

**Decision letter after peer review:**

Thank you for submitting your article "Structure of Dunaliella Photosystem II reveals conformational flexibility of stacked and unstacked supercomplexes" for consideration by *eLife*. Your article has been reviewed by 3 peer reviewers, and the evaluation has been overseen by a Reviewing Editor and Benoît Kornmann as the Senior Editor. The following individual involved in the review of your submission has agreed to reveal their identity: Helmut Kirchhoff (Reviewer #1).

Essential revisions:

The reviewers were enthusiastic about the quality of the data and its potential implications. Nevertheless, there were some concerns that need to be addressed, though these will probably not require additional experimentation.

The authors should take extra care to explain what limits of their interpretations, and in particular be completely open about any possible artifacts introduced by the isolation and imaging in cryoEM. Keep in mind that many readers will have doubts about these kinds of artifacts, and it is better to both explain these potential issues, erring on the side of caution.

*Reviewer #1 (Recommendations for the authors):*

I have no concerns about the quality of the work, how the authors interpreted the data, and what they concluded about the structural organization of PSII with one exception. The authors, however, should consider that the two PSII structures and the 'stacked PSII' sandwich could be an artificial result of the isolation procedure. A critical discussion of this aspect would increase the credibility of the work. For example, there is quantitative information available on the stacking of thylakoid membranes in *Chlamydomonas* and higher plants, i.e. the distance between membranes in appressed grana regions. It would be very interesting to back-project the structure of the stacked PSII sandwich into a pair of stacked membranes found in native cells. The latter can be taken from the literature. Does the PSII sandwich predict the membrane-membrane separation for native membranes? That would add a very interesting aspect to this already compelling story.

*Reviewer #3 (Recommendations for the authors):*

Overall, this is a well-written manuscript with high-quality cryo-EM data processing, maps, and models. Below are several suggestions that I think would strengthen the overall arguments.

1. Supplemental figure 1C should be expanded to show all classes from all rounds of 3D classification, and clearly delineate when symmetry was or was not applied at each individual step of classification/refinement.

2. The ability to resolve water molecules is impressive, and the authors should capitalize on this opportunity to demonstrate the quality of their high-resolution reconstruction. The addition of a panel in Supplemental figure 3 clearly showing density for multiple waters would go a long way in demonstrating the map quality.

3. Please ensure that the maps deposited to the EMDB are in an easily usable format for non-expert users of molecular visualization tools (ie: UCSF Chimera). In order to analyze the maps provided during the review I had to convert them out of float16 format in order to open them in Chimera, and at least for one of the maps I had to z-flip in order to invert the map handedness to match the atomic model. For people who are not intimately familiar with such processes, it would be very difficult to analyze the models/maps.

4. Please consider depositing the individual maps for bodies from multibody refinement, in addition to the map of combined bodies. This allows future users to combine maps in alternate fashions if they choose.

5. In Supplemental table 1, please consider providing a b-factor for water separate from ligands.

6. Supplemental table 1, there seems to be an error in "Model composition". There are only ~50-100 protein residues listed.

---

## [Author Response]

Reviewer #1 (Recommendations for the authors):I have no concerns about the quality of the work, how the authors interpreted the data, and what they concluded about the structural organization of PSII with one exception. The authors, however, should consider that the two PSII structures and the 'stacked PSII' sandwich could be an artificial result of the isolation procedure. A critical discussion of this aspect would increase the credibility of the work. For example, there is quantitative information available on the stacking of thylakoid membranes in Chlamydomonas and higher plants, i.e. the distance between membranes in appressed grana regions. It would be very interesting to back-project the structure of the stacked PSII sandwich into a pair of stacked membranes found in native cells. The latter can be taken from the literature. Does the PSII sandwich predict the membrane-membrane separation for native membranes? That would add a very interesting aspect to this already compelling story.

We do not dispute that preparation conditions affect the formation of the stacked PSII dimer and this is noted in our manuscript, however, this does not imply that it is not biologically relevant. Several PSII stacked complexes were previously identified in vitrified thylakoid grana stacks using cryo-electron tomography. We selected a recent tomogram by *Wietrzynski et al.* as a reference. Although *Wietrzynski et al.* present evidence to the fact that stacked PSII dimers are rarely observed in vivo, this is one of the best examples of the in-vivo dimensions of the stacked PSII. Back projecting the stacked PSII into *Chlamydomonas* native cell tomograms shows an excellent alignment in all available features, including the stromal extrinsic subunits and the intermembrane space. This supports the similarity between the cryo-EM observed stacked PSII and an in-vivo stacked PSII configuration.

Our suggestion is simply that stacked PSII dimers can play a role in supporting grana structure and that the large flexibility we observed in PSII orientations can lead to under-detection of these dimers in vivo.

We added the definition for state 1 and state 10 in the legend of Figure 2.

Reviewer #3 (Recommendations for the authors):Overall, this is a well-written manuscript with high-quality cryo-EM data processing, maps, and models. Below are several suggestions that I think would strengthen the overall arguments.1. Supplemental figure 1C should be expanded to show all classes from all rounds of 3D classification, and clearly delineate when symmetry was or was not applied at each individual step of classification/refinement.

We have added these details to Figure 1—figure supplement 1 and they are now clearly indicated.

2. The ability to resolve water molecules is impressive, and the authors should capitalize on this opportunity to demonstrate the quality of their high-resolution reconstruction. The addition of a panel in Supplemental figure 3 clearly showing density for multiple waters would go a long way in demonstrating the map quality.

We have added panel d to Figure 1—figure supplement 3 to show all the water molecules coordinated by CP43 with their map densities.

3. Please ensure that the maps deposited to the EMDB are in an easily usable format for non-expert users of molecular visualization tools (ie: UCSF Chimera). In order to analyze the maps provided during the review I had to convert them out of float16 format in order to open them in Chimera, and at least for one of the maps I had to z-flip in order to invert the map handedness to match the atomic model. For people who are not intimately familiar with such processes, it would be very difficult to analyze the models/maps.

Conversion to float16 was done solely to reduce the file size in order to upload the maps. The maps deposited on the EMDB are at the correct orientation and in regular MRC format.

4. Please consider depositing the individual maps for bodies from multibody refinement, in addition to the map of combined bodies. This allows future users to combine maps in alternate fashions if they choose.

We prefer to deposit only a single map to avoid confusion although we acknowledge that this results in some data loss.

5. In Supplemental table 1, please consider providing a b-factor for water separate from ligands.

Done.

6. Supplemental table 1, there seems to be an error in "Model composition". There are only ~50-100 protein residues listed.

This entry mistakenly listed chains numbers, we corrected it to the show protein residues.